# Hierarchical Linear Model of Internet Addiction and Associated Risk Factors in Chinese Adolescents: A Longitudinal Study

**DOI:** 10.3390/ijerph192114008

**Published:** 2022-10-27

**Authors:** Guangming Li

**Affiliations:** 1Key Laboratory of Brain, Cognition and Education Sciences, Ministry of Education, South China Normal University, Guangzhou 510631, China; lgm2004100@m.scnu.edu.cn; 2School of Psychology, Center for Studies of Psychological Application, Guangdong Key Laboratory of Mental Health and Cognitive Science, South China Normal University, Guangzhou 510631, China

**Keywords:** Chinese adolescents, internet addiction, associated risk factors, hierarchical linear model, longitudinal analysis

## Abstract

The risk effects of internet addiction have been documented in the literature; however, few longitudinal studies have considered the heterogeneity of the subjects. A hierarchical linear model was used here to explore the relationship between adolescents’ internet addiction and associated risk factors (depression, anxiety, gender, and obesity) from the perspective of longitudinal analysis. A total of 1033 adolescents were investigated and followed up with every three months with the Self-Rating Depression Scale (SDS), Self-Rating Anxiety Scale (SAS), and Internet Addiction Test (IAT). The hierarchical linear model of internet addiction had only two levels. The first level of the model was the time variable (three time points) and the second level of the model was the individual adolescent (1033 adolescents). The results showed that (1) depression and anxiety, as associated risk factors, were significant positive predictors of adolescents’ internet addiction considering the developmental trajectory courses of adolescent internet addiction, as well as the individual differences over time; (2) there were gender differences in the adolescents’ internet addictions—specifically, the initial level of internet addiction among boys was significantly higher than that of girls, but the rate of decline was significantly faster than that of girls; and (3) there was no significant difference in obesity. The results demonstrated the importance of considering depression, anxiety, and gender in any intervention efforts to reduce adolescents’ internet addictions, and we should pay attention to the cultivation of positive coping strategies for Chinese adolescents. The limitations of the study were also discussed.

## 1. Introduction

With the progress of science and technology, and the rapid development of computers and mobile phones, the internet has become an indispensable part of people’s lives. The internet has become a necessary resource for adolescents to receive and share new knowledge and information [1]. For adolescents who easily accept new things, social media networks have become one of the main sources of information they receive [2,3]. However, there will be many problems when adolescents enjoy the convenience brought by the internet. Some studies have found that the excessive use of the internet will affect our lives, study, work, and both physical and mental health [4,5,6]. Some scholars call the previously mentioned phenomenon “ internet addiction”.

The concept of internet addiction was first proposed by Goldberg, also known as pathological network use, internet dependence, etc. [7,8,9,10,11,12,13]. Although the terms used to describe internet addiction are different, their connotations are basically the same, mainly referring to individuals’ unrestrained internet use, which also has a negative impact on their lives. Nowadays, internet addiction has become a worldwide problem attracting much attention from psychologists [14,15,16]. Previous studies have found that internet addiction is an associated risk with a variety of psychological problems, such as depression [17,18,19], insomnia [20], anxiety [21,22], attention deficit and hyperactivity disorder [23], loneliness [24], aggressive behavior [25], and so on, among which the relationships between depression, anxiety, and internet addiction are particularly prominent in adolescents [26].

The Mood Enhancement Hypothesis proposed by [27] suggests that individuals with negative emotions are more likely to seek entertainment to relieve their stress. According to the theory, adolescents’ depression and anxiety are more likely to be alleviated through entertainment activities in a virtual world (such as watching TV, surfing the internet, and playing online games), and when they are exposed to virtual worlds for too long, these adolescents are more likely to become addicted to the internet. In fact, many studies have confirmed the hypothesis above that there is a significant positive relationship between depression, anxiety, and internet addiction. The higher the level of depression and anxiety, the more serious the internet addiction [13,28]. However, most studies were conducted from a cross-sectional perspective, which provided a lot of valuable information, but could not reflect the changes in internet addiction, depression, and anxiety over time. Therefore, this study explored the relationship between depression, anxiety, and internet addiction from a longitudinal perspective, with the hypothesis that depression and anxiety can significantly and positively predict internet addiction over time.

In addition to the above discussion of the relationship between depression, anxiety, and internet addiction, this study also considered other associated risk factors (gender and obesity) to explore whether there are significant differences in gender and obesity as variables of internet addiction.

It was found that the discussions of whether there is a gender difference in internet addiction were mostly carried out from the perspective of cross-sectional analysis, which reached a relatively consistent conclusion that there is a significant gender difference in internet addiction, and boys are significantly more addicted to the internet than girls [29,30]. However, as time goes on, adolescents’ internet addiction will undergo a dynamic development process. Is there also a gender difference in the change trend of internet addiction? This problem cannot be solved in cross-sectional studies. Therefore, it is necessary to explore whether there is a gender difference in the change trend of internet addiction from the perspective of longitudinal analysis. Although the above questions have not been answered in the internet addiction field, Young and Rogers [13] pointed out that internet addiction is a kind of behavior, like pathological gambling. Studies in the field of pathological gambling have found that women are worse at gambling than men [31,32].

Therefore, the second hypothesis of this study is that men are more likely to develop internet addiction than women, but women’s internet addiction will worsen faster than that of men.

Obesity is the second associated risk factor included in this study. In fact, obesity has become a global problem, and it is getting worse in both developed and developing countries [33,34,35,36,37]. Previous studies have shown that obesity is closely related to internet use [33,38], and individuals who use the internet for a long time are more likely to be obese [39,40,41]. However, there are some studies directly discussing internet addiction and obesity among adolescents [39,42]. However, it is not clear whether there is a significant difference between the obese group and non-obese group from a longitudinal perspective. Therefore, the third task of this study is to explore whether there is a significant difference in the change trend of internet addiction between the obese group and the non-obese group. Moreover, we hypothesize that the obese group will be more likely to develop internet addiction than the non-obese group, and the obese group’s internet addiction will worsen faster than that of the non-obese group.

To verify the third hypothesis, this study uses a hierarchical linear model to explore adolescents’ internet addiction from the perspective of longitudinal research. In the hierarchical linear model, first, adolescents’ internet addiction is taken as a dependent variable, and the first-level regression equation was established with time-varying variables (such as depression and anxiety) as independent variables. Then, a second-level regression equation is established using the intercept and slope of the first-level model as the dependent variables. Therefore, the hierarchical linear model is also known as the "return of the return". As a common method of longitudinal analysis, hierarchical linear models can not only reasonably reflect the causality among variables, but also describe the differences among individuals.

To sum up, the purposes of this study are as follows: (1) to explore the longitudinal effects of depression and anxiety on adolescents’ internet addiction; (2) to explore whether there is a significant difference based on gender for the changing trend of adolescents’ internet addiction; and (3) to explore whether there is a significant difference in obesity variables in the change trend of adolescents’ internet addiction.

## 2. Methods

### 2.1. Participants

The participants were a sample of Chinese adolescents (N = 1405, 52.0% girls) between 11 and 19 years of age (M = 14.58; SD = 1.91) from three middle schools (17 classes) in an urban school district in South China. Of these, 1033 adolescents (73.5%; 419 boys and 614 girls) provided complete data at both Time 2 and Time 3. A proportion of 15.9% of the adolescents were measured as obese. Their basic information is shown in Table 1. We conducted a series of missing analyses and the results showed that, at baseline (T1), adolescents who had complete data were more likely than those who had missing data to be girls, *χ*^2^(1) = 12.035, *p* < 0.001, and to report a higher level of depression, *t*(1521) = −2.461, *p* = 0.015. There were no differences in IA, anxiety, obesity, and age.

In this study, a total of 1033 middle school students were selected. Different classes had different students, and each class had about 60 students. They were contacted every three months for two follow-up tests (T2 and T3). As shown in Table 1, of the 1033 middle school students, 419 were male (40.6%) and 614 were female (59.4%); 452 were aged from 11 to 15 and 581 were aged from 16 to 19; 171 were obese (16.6%) and 862 were non-obese (83.4%); The average age of boys was 15, and that of girls was 14.8.

### 2.2. Procedure

We randomly selected three middle schools in urban areas in southern China, and then randomly chose 17 classes from these three middle schools. Different students were nested in different classes. Data were collected using paper/pencil tests in classrooms every three months for a total study span of six months. Each participant had a unique ID, and the same ID was used at all three time points. No participants changed classes or schools during these six months. The measurement sites were provided by the Academic Affairs Offices of the three middle schools. The researchers were trained before they administered the survey. Both schools and parents agreed to the assessment, and this research was approved by the South China Normal University (SCNU) research ethics board (Institutional Review Board).

### 2.3. Measures

#### 2.3.1. Internet Addiction

The measurement of adolescents’ internet addiction is generally based on the Internet Addiction Test (IAT) developed by [11], the Chinese version of which consists of 20 questions. Some items are about the usage time. Each topic is scored from 1 to 5 points (from 1 = “very little” to 5 = “always”), so the minimum and maximum test scores are 20 and 100, respectively. The higher the score, the more serious the internet addiction behavior. Previous studies have shown that IAT has high reliability and validity, and it is an effective tool to measure internet addiction behavior [7,43,44]. The internal consistency (Cronbach’s alpha) for the Internet addiction items was good (T1 α = 0.91; T2 α = 0.92; and T3 α = 0.93).

#### 2.3.2. Anxiety

Adolescents’ anxiety was measured by using the Self-rating Anxiety Scale (SAS) proposed by [45]. The SAS contains a total of 20 questions graded on the four-point Likert scale (1 = “rarely”, 4 = “always”). Of these 20 questions, 5 questions (5, 9, 13, 17, and 19) require a reverse scoring. The highest and lowest scores of the test are 80 and 20, respectively, and the higher the score, the greater the anxiety. In fact, the SAS has been widely used in the assessment of anxiety and has shown good reliability and validity [2,46,47,48]. The internal consistency (Cronbach’s alpha) for the anxiety items was good at each time point (T1 α = 0.76; T2 α = 0.79; and T3 α = 0.74).

#### 2.3.3. Depression

Adolescents’ depression was measured by using the Self-rating Depression Scale (SDS) proposed by [49]. The SDS consists of 20 questions, each of which is scored on a scale of 1 to 4 (1 = “rarely”, 4 = “always”). Higher SDS scores indicate a higher level of depression. Among the 20 questions, there are 10 reverse scoring questions, which need to be reversed in future calculations. Previous studies have shown that the SDS has good reliability and validity [50,51]. The internal consistency (Cronbach’s alpha) of the depression items was good (T1 α = 0.79; T2 α = 0.83; T3 α = 0.85).

#### 2.3.4. Other Associated Risk Variables

This study included two other associated risk variables, gender and obesity. The gender information was obtained by self-reporting in the initial test. In this study, gender was coded as 0 or 1, 0 for men and 1 for women. Obesity was computed by calculating the body mass index (BMI) based on height and weight, following the Group of China Obesity Task Force (2004), and the adolescents were coded as 0 (no-obese) or 1 (obese) [1].

### 2.4. Data Analysis

In this study, SPSS 19.0 was used to conduct descriptive statistical analysis of each variable first to gain a general understanding of adolescents’ internet addictions. Then, HLM6.0 was used to analyze the zero model of the hierarchical linear model, exploring whether there were differences in adolescents’ internet addiction and whether it was necessary to stratify it. Finally, HLM6.0 was used again to analyze the complete model of the hierarchical linear model and explore the longitudinal effects of the variables (anxiety, depression, gender, and obesity) included on adolescents’ internet addiction.

## 3. Results

### 3.1. Descriptive Statistical Analysis

To gain a preliminary understanding of adolescents’ internet addiction, anxiety, depression, gender, and obesity, this study conducted a simple descriptive statistical analysis first, as Table 2 shows.

### 3.2. Effects of Depression and Anxiety on Adolescents’ Internet Addiction

To gain a scientific understanding of adolescents’ internet addiction, this study used the hierarchical linear model to analyze the data. Firstly, the null model was established to analyze whether it was necessary to establish a hierarchical linear model. Secondly, based on the analysis results of the null model, the predicted variables (depression and anxiety) that changed over time and demographic variables (obesity and gender) were included to establish the complete model, and the longitudinal effects of various factors (depression, anxiety, obesity, and gender) on adolescents’ internet addiction were further analyzed.

#### 3.2.1. Null Model

The null model is mainly used to test whether the growth trend of individual variables is different, which is the basis of the later complete model. In this study, a null model was established with adolescents’ internet addiction as the dependent variable. The equation of the first level of the model and the equation of the second level are as follows:IA = B0 + R   B0 = G00 + U0(1)
where B0 represents the intercept of the first level of the model, G00 represents the intercept of the second level of the model, and R and U0 represent the residual terms of the first and second levels of the model, respectively. The results of the null model are shown in Table 3.

Firstly, the parameter estimation results of the fixed part indicated that the overall mean of the three measurements of internet addiction was 31.60. Secondly, the parameter estimation results of the random part showed that there were significant differences between individuals in terms of internet addiction (variance = 70.58, *χ*^2^ = 5065.59, *df* = 1032, *p* < 0.001). The cross-level correlation results showed that ICC = 70.58/(70.58 + 54.17) = 56.58%, indicating that 56.58% of the variation in adolescents’ internet addiction was explained by the subject’s internal variables, illustrating the necessity and rationality of establishing a hierarchical linear model.

#### 3.2.2. Complete Model

Based on the results of the null model, this study took internet addiction as the dependent variable, establishing the regression equation incorporating the predicted variables varying over time in the first-level equation, such as the times coded 0, 1, and 2, depression, and anxiety. In the second-level equation, the intercept and the slope of time of the first-level equation were taken as the dependent variable, which established a complete model of the regression equation including demographic variables, such as gender (0 = male and 1 = female) and obesity (0 = non-obese group and 1 = obese group), to analyze the longitudinal effects of adolescents’ internet addiction that anxiety, depression, gender, and obesity have caused.

The sample design indicates that different times were clustered in individual students (developmental model). The hierarchical linear model of internet addiction has only two levels. The first level of the model is the time variable (three time points) and the second level of the model is the individual adolescent (1033 adolescents). The equations of the first level and the second level of the model for the complete model are as follows:         internet addiction = B0 + B1*(time) + B2*(anxiety) + B3*(depression) + R                    B0 = G00 + G01*(obesity) + G02*(gender) + U0                    B1 = G10 + G11*(obesity) + G12*(gender) + U1  B2 = G20  B3 = G30(2)
where B0 represents the intercept of the first level of the model, B1 represents the slope of time for the first level of the model, B2 represents the slope of anxiety for the first level of the model, B3 represents the slope of depression for the first level of the model, G00 represents the intercept of the second level of the model, G01 represents the slope of obesity for the second level of the model, G02 represents the slope of obesity for the second level of the model, etc. R and U0 represent the residual terms of the first and second levels of the model, respectively. 

According to Equation (2), there are were multidisciplinary effects (tolerance = 0.59, VIF = 1.69, and DW = 1.36). The specific analysis results of the complete model are shown in Table 4. Firstly, it can be seen from the intercept estimation results of the first-level model that the mean initial internet addiction in the non-obese men group was 35.46, and there was a significant difference between men and women in the initial condition of internet addiction (*p* < 0.001). Firstly, boys’ initial internet addiction was significantly higher than that of girls, but there was no significant difference between the obese group and the non-obese group (*p* > 0.05). Secondly, it can be seen from the estimation results of the time slope of the first-level model that the worsening rate of internet addiction in boys was significantly slower than that in girls, but there was also no significant difference between the obese group and non-obese group (*p* > 0.05). Finally, the parameter estimation results of anxiety and depression showed that both anxiety and depression could positively predict adolescents’ internet addiction.

The complete model not only estimated the fixed parameters, but also the random parameters. According to the results in Table 4, no matter at initial level (level 2 = 3191.10, *df* = 1030, *p* < 0.001) or the rate of change (level 2 =1117.13, *df* = 1030, *p* < 0.05), there were still individual differences in adolescents’ internet addiction after the inclusion of the variables of gender and obesity.

## 4. Discussion

With the development of science and technology, the internet has become an indispensable part of people’s lives. Adolescents who are open to new things are more likely to be interested in the internet. However, lacking self-control, adolescents will be affected by abundant recreational activities, which will lead to addiction and eventually internet addiction. Previous studies have found that internet addiction not only affects individuals’ work and study, but also causes various psychological diseases [15,17,42]. Therefore, it is necessary to discuss adolescents’ internet addiction. Differing from the studies on internet addiction from the perspective of cross-sectional analysis, this study adopts a hierarchical linear model to explore the internet addiction of adolescents from the perspective of longitudinal analysis, investigating the vertical influences of depression, anxiety, gender, and obesity on adolescents’ internet addiction.

### 4.1. Effects of Depression and Anxiety on Adolescents’ Internet Addiction

The results in Table 4 found that both depression and anxiety positively predicted adolescents’ internet addiction; that is, the higher the levels of depression and anxiety, the more serious adolescents’ internet addiction. The above results confirmed the first hypothesis of this study.

The Mood Enhancement Hypothesis proposed by [27] explains the above results, suggesting that individuals with negative emotions are more likely to release their stress through entertainment activities. That is, adolescents with depression or anxiety are more likely to alleviate their depression and anxiety through entertainment activities in the virtual world. However, adolescents with depression or anxiety are more likely to become addicted to the internet when they are exposed to the entertainment activities in the virtual world for too long. Furthermore, previous studies have found that adolescents with depression and anxiety often have difficulties in real communication. They often feel frustrated in real-life communication [32,52]. Negative feelings will undoubtedly prompt adolescents to avoid real-life communication. Because of the concealment, convenience, and anonymity of the network, adolescents who are frustrated in real-life interactions will actively seek online communication, and long online interactions will gradually make them lose skills and abilities in realistic communication [53]. Repeatedly, adolescents will fall into a vicious circle of circles and eventually become addicted to the internet.

The prevention or treatment of adolescents’ internet addiction should be carried out from the following aspects: on the one hand, we should supervise adolescents’ usage time of the internet. There is no doubt that the irrational use of the internet is the direct cause of adolescents’ internet addiction. Smahel et al. [54] study found that the time spent online is positively correlated with internet addiction; that is, the longer the time spent online, the more serious the internet addiction. Due to adolescents’ relatively weak self-control, the abundant entertainment activities on the internet will make adolescents indulge in it for a long time; therefore, parents and teachers have the responsibility to do a good job of supervision, limiting adolescents’ internet usage time. On the other hand, adolescents’ communication skills and abilities in real life can be improved. It is not only necessary to control adolescents’ usage time of the internet only, but to also find the deeper causes of internet addiction and take measures to prevent or treat it. Studies have found that failure in real-life communication is the main reason for adolescents to seek online communication [55]. Therefore, it is necessary to improve adolescents’ practical communication skills and abilities and enhance their positive expectations for real communication through a series of training and learning activities. Finally, attempts to reduce adolescents’ negative emotions (such as depression and anxiety) should be made. The results of this study show that negative emotions, such as depression and anxiety, are the reasons for the acceleration of adolescents’ internet addiction. Therefore, in the process of the prevention and treatment of adolescents’ internet addiction, some courses for adjusting depression and anxiety should be brought in to control adolescents’ negative emotions.

### 4.2. Performance of Adolescents’ Internet Addiction within the Gender Variable

According to Table 4, there is a significant gender difference in adolescents’ internet addiction; specifically, the initial level of internet addiction among boys was significantly higher than that of girls, but the rate of decline was significantly faster than that of girls. The above results confirmed the second hypothesis of this study.

In fact, gender is a very important variable in the discussion of adolescents’ internet addiction. Previous studies also consistently found that internet addiction among boys is significantly higher than that among girls [28,29,30], which was explained by different scholars from different perspectives. Firstly, some scholars believe that, due to their time spent online, internet addiction among boys is significantly higher than that among girls. Studies have found that boys spend significantly more time on the internet than girls [52,56], and internet usage time is positively correlated with internet addiction; that is, the longer the internet usage time, the more serious the internet addiction. Therefore, as boys spend more time on the internet than girls, their internet addiction is significantly more serious than that of girls. Secondly, some scholars believe that, due to different coping strategies, boys’ internet addiction is significantly higher [3,6]. Previous studies found that boys are more likely to use avoidance strategies than girls [1,57]. When boys encounter setbacks and stress in life, they usually avoid stressful events, rather than seek help to solve the problem actively. Therefore, boys are more likely to avoid stressful events via the internet and are more likely to have higher levels of internet addiction than girls. This study also further found that the rate of deterioration of internet addiction among boys was significantly slower, which was consistent with the results of studies in the field of pathological gambling. The rapid deterioration of internet addiction among girls is also known as the “Stretching Effect” [31], which has been explained by researchers based on the biological vulnerability of females.

The results above reveal that, in the intervention process for adolescents’ internet addiction, gender differences must be considered and targeted intervention measures should also be taken. Research in the future needs to explore the causes of significant gender differences in adolescents’ internet addiction.

### 4.3. Performance of Adolescents’ Internet Addiction within the Obesity Variable

Table 4 shows that there was no significant difference in adolescents’ internet addiction for the obesity variable. This is inconsistent with the second hypothesis of this study. The results can be explained by the different categories of internet use (e.g., watching TV, playing digital games, etc.).

Existing studies have found a positive correlation between watching TV, using computers (such as browsing websites, sending emails, etc.), and obesity, but no significant correlation between playing digital games (such as video games, computers, and game consoles) and obesity [2,39]. In other words, the relationship between internet use and obesity varies between different categories of internet use. Therefore, different categories of internet use can explain the inconsistency in the relationship between internet addiction and obesity. Studies in the future can also classify internet addiction into different categories to further explore the relationship between internet addiction and obesity.

### 4.4. Shortages and Future Work

Although adolescents’ internet addiction was studied from the longitudinal perspective and the effects of depression, anxiety, and gender on adolescents’ internet addiction were also fully discussed, there are still some shortages in this work. Firstly, only the score of IAT was considered regarding internet addiction. There was no consideration of other contents, such as games, social media, YouTube, etc. Internet addiction, depression, and anxiety in this paper only represent the extent of internet addiction, depression, and anxiety, which is a continuum and is not classified by them. Secondly, this study only included four variables, i.e., depression, anxiety, gender, and obesity. After the inclusion of these variables, there were still individual differences in internet addiction at both the initial level and for the rate of change. Therefore, it is necessary to consider more factors to discuss adolescents’ internet addiction in the future, such as parental education, improving the research. Finally, the data of this study were obtained from the subjective responses of the subjects, lacking an objective test. In the future, some objective tests could be added to test the subjects.

## 5. Conclusions

This study uses a hierarchical linear model to explore the relationship between adolescents’ internet addiction and depression, anxiety, gender, and obesity from the perspective of longitudinal analysis. Few published studies have examined these relationships considering the hierarchical heterogeneity of participants, as well as the specific patterns of internet addiction. This study makes some conclusions, as follows:

(1) Higher depression and anxiety, as associated risk factors, are factors that hinder the decline in adolescents’ internet addiction considering the developmental trajectory courses of adolescents’ Internet addiction as well as the individual differences over time.

(2) There is a significant gender difference in adolescents’ internet addiction. Specifically, the initial level of internet addiction among boys was significantly higher than that of girls, but over time, boys’ internet addiction decreased significantly faster than that of girls. 

(3) There is no significant difference in adolescents’ internet addiction based on obesity. The relationship between internet use and obesity varies with different categories of internet use.

## Figures and Tables

**Table 1 ijerph-19-14008-t001:** Basic information of the subjects.

	*n*	%
Gender		
Male	419	40.6
Female	614	59.4
Age		
11–15	452	43.8
16–19	581	56.2
Obesity		
Obese	862	83.4
Non-obese	171	16.6

**Table 2 ijerph-19-14008-t002:** Mean value, standard deviation, and correlation coefficient of each variable.

	M	SD	1	2	3	4	5	6	7	8	9	10	11
1 Obesity	0.17	0.37	1										
2 Gender	0.59	0.49	0.05	1									
3 Internet addiction T1	33.59	12.01	−0.03	−0.08 **	1								
4 Internet addiction T2	31.67	10.83	−0.02	−0.09 **	0.58 **	1							
5 Internet addiction T3	29.52	10.23	−0.04	−0.02	0.61 **	0.59 **	1						
6 Anxiety T1	31.59	6.76	0.06	0.19 **	0.19 **	0.09 **	0.17 **	1					
7 Anxiety T2	31.98	7.17	0.01	0.05	0.24 **	0.41 **	0.28 **	0.21 **	1				
8 Anxiety T3	31.28	6.70	0.02	−0.01	0.25 **	0.21 **	0.26 **	0.12 **	0.41 **	1			
9 Depression T1	39.37	8.24	0.06	0.16 **	0.11 **	0.03	0.10 **	0.56 **	0.02	−0.02	1		
10 Depression T2	36.71	8.67	0.01	−0.02	0.24 **	0.34 **	0.24 **	0.06 **	0.67 **	0.44 **	−0.16 **	1	
11 Depression T3	38.86	9.72	−0.01	0.21 **	−0.06	−0.07 *	0.01	0.22 **	−0.14 **	−0.27 **	0.34 **	−0.32 **	1

Note: * *p* < 0.05, ** *p* < 0.01.

**Table 3 ijerph-19-14008-t003:** Analysis results of the null model.

Fixed Parameter		*β*	*SE*	*p*
Mean (G00)		31.60	0.29	*p* < 0.001
Random parameter	Variance component	*χ* ^2^	*df*	*p*
Inter-subject residual (U0)	70.58	5065.59	1032	*p* < 0.001
Within-subject residual (R)	54.17			

**Table 4 ijerph-19-14008-t004:** Analysis results of the complete model.

Fixed Parameter		*β*	*SE*	*p*
Intercept estimates of the first-level model (B0)				
Mean G00		35.46	0.60	*p* < 0.001
Obesity G01		−1.00	0.94	*p* > 0.05
Gender G02		−2.89	0.73	*p* < 0.001
Time slope estimates of the first-level model (B1)				
Mean G10		−2.62	0.28	*p* < 0.001
Obesity G11		0.08	0.41	*p* > 0.05
Gender G12		1.05	0.33	*p* < 0.01
Estimates of anxiety slope (B2)				
Mean G20		1.80	0.22	*p* < 0.001
Estimates of depression slope (B3)				
Mean G30		0.58	0.15	*p* < 0.001
Random parameter	Variance component	*χ* ^2^	*df*	*p*
Inter-subject residual (U0)	82.18	3191.10	1030	*p* < 0.001
Inter-subject residual (U1)	2.24	1117.13	1030	*p* < 0.05
Within-subject residual (R)	46.87			

## Data Availability

The data are not publicly available due to privacy restrictions.

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
