# Peer review of "Hierarchical Linear Model of Internet Addiction and Associated Risk Factors in Chinese Adolescents: A Longitudinal Study"

_ijerph, 2022, doi:10.3390/ijerph192114008_

Round 1
Reviewer 1 Report (Previous Reviewer 1)
The authors revised their manuscript based on our suggestions.
Author Response
thank you.
Reviewer 2 Report (Previous Reviewer 2)
This revised version can be published.
Author Response
thank you.
Reviewer 3 Report (Previous Reviewer 3)
There continue to be issues with the manuscript that I raised previously and were not addressed:
- Numerous instances of inappropriate word choice (e.g., “horizontal analysis”)
- Many grammatical errors (e.g., missing or misspelled words – please do a search on addition when you mean addiction)
- I previously asked about parental education. The response was that this was “significant and in not considered for subsequent analysis”. Why not?
My primary concern previously was not addressed at all with the revisions, nor in the response to the reviewers. Specifically, I raised the issue that this is a sample of adolescents who are typical internet users and no evidence of internet addiction. I requested the proportion of students meeting the criteria of internet addiction and that was not provided. I also requested the proportion of students that met the criteria for being depressed or anxious and these were also not included.
Author Response
There continue to be issues with the manuscript that I raised previously and were not addressed:
- Numerous instances of inappropriate word choice (e.g., “horizontal analysis”)
Answer: Thanks for your comments! The horizontal analysis has been modified to cross-sectional analysis. The horizontal study has been modified to cross-sectional study. The horizontal perspective has been modified to cross-sectional perspective.
- Many grammatical errors (e.g., missing or misspelled words – please do a search on addition when you mean addiction)
Answer: Thanks for your comments! The addition has been modified to addiction. The “In addition” has been modified to “Moreover” or “Besides” or “Furthermore”. The Internet has been modified to internet. We also checked some other problems about grammatical errors.
- I previously asked about parental education. The response was that this was “significant and in not considered for subsequent analysis”. Why not?
Answer: Thanks for your comments! I am very sorry. I write it wrongly. The following answer is correct:
In fact, we have a measure of parental education, but, it is not significant and is not considered for subsequent analysis.
I am very sorry!
For this reason, we add the following explanations to the research limitations:
Therefore, it is necessary to consider more factors to discuss adolescents’ internet addiction in the future such as parental education, making the research better.
Please see the 4.4. Shortage and Envisage.
- My primary concern previously was not addressed at all with the revisions, nor in the response to the reviewers. Specifically, I raised the issue that this is a sample of adolescents who are typical internet users and no evidence of internet addiction. I requested the proportion of students meeting the criteria of internet addiction and that was not provided. I also requested the proportion of students that met the criteria for being depressed or anxious and these were also not included.
Answer: Thanks for your comments! The internet addiction in this paper does not mean that there must be internet addiction. The internet addiction in this paper only represents the extent of internet addiction, which is a continuum, not classified by it. That is to say, we did not use the Internet Addiction Test (IAT) scale to distinguish between those who are internet addiction groups and those who are not, but only to indicate the extent of internet addiction. In fact, so are anxiety and depression. Of course, this statement has certain limitations. For this reason, we add the following explanations to the research limitations:
The internet addiction, depression and anxiety in this paper only represent the extent of internet addiction, depression and anxiety, which is a continuum, not classified by them.
Please see the 4.4. Shortage and Envisage.
Sorry, the previous version didn't explain it clearly, thank you!

Round 2
Reviewer 3 Report (Previous Reviewer 3)
Thank you for addressing the issues I raised in the previous review. The version of the manuscript provided to me to review has dropped the additional authors who were added in the previous version. What is the rationale for this change?
Author Response
Thank you for addressing the issues I raised in the previous review. The version of the manuscript provided to me to review has dropped the additional authors who were added in the previous version. What is the rationale for this change?
Answer:
Thanks for your comments!
Another author did not have an official email address and was unwilling to modify it or provide the official email address of the school. I communicated with her many times. She thought it was too troublesome and gave up. I have reported this matter to the editorial department or chief editor of IJERPH, and explained this change by telephone.
Thank you. Best wish to you.
This manuscript is a resubmission of an earlier submission. The following is a list of the peer review reports and author responses from that submission.
Round 1
Reviewer 1 Report
Thank you for the invite to review the manuscript “Longitudinal Analysis of Adolescents’ Internet Addition Based on Hierarchical Linear Model”. This survey examined the relationship between internet addiction and depression, anxiety, gender and obesity among adolescents.
I have a few points of concern, I recommend to edit the manuscript.
1. The title is "Longitudinal Analysis of Adolescents’ Internet Addition Based on Hierarchical Linear Model". It is difficult to understand the type of study design, what is clinical sample or cohort, in this manuscript for reader. Therefore, it is need to change title.
2. The abstract lacks an introduction and discussion.
3. Author mentioned “We randomly selected three middle schools in urban areas in southern China, and then randomly chose 17 classes from these three middle schools.”. Author showed the methods of sampling. However, the flow of 1033 sample are unknown, for example the number of all students, exclusion criteria. In addition, the calculation of sample size methods is necessary.
4. The criteria for determining whether subjects are obese are based on the Chinese classification standards of overweight and obese BMI (China working group on obesity, 2004). However, almost students were matched the criteria of obesity. Thus, the criteria is may be not suitable for students. In addition, the detail of obese BMI is unknown.
5. The age of participants is not clear.
6. Some variables may have issues of multicollinearity. Please provide more information whether it met the requirement to be included in the analysis.
7. Only the score of IAT is considered regarding internet addiction. No consideration of another contents, such as games, social media, and YouTube, etc. In addition, it is not considered about the using time. There are major limitations.
Reviewer 2 Report
The manuscript is poorly written and edited, beginning from the title: ‘Longitudinal Analysis of Adolescents’ Internet Addition Based on Hierarchical Linear Model’ – Addiction, not Addition. And be restricted on China. ‘computer network, network’ - – avoid close repetitions. ‘accept new things, network has’ – vague, you mean ‘social media networks’. ‘Ko, Yen, Chen, Yeh, & Yen, 2009’ – correct in-text citation: Ko et al., 2009. Check throughout the manuscript. ‘However, everything has two sides.’ – vague. The proportion of old sources is extremely high. ‘which attracts much attention of psychologists’ – not only. Inconsistent citation style: ‘The Mood Enhancement hypothesis proposed by Bryant and Zillmann suggests that individuals with negative emotions are more likely to seek entertainment to relieve their stress (Bryant & Zillmann, 1984). – should be ‘The Mood Enhancement hypothesis proposed by Bryant and Zillmann (1984) suggests that individuals with negative emotions are more likely to seek entertainment to relieve their stress.’ The same here: ‘Although the questions above were not answered in the internet addiction field, Young and Rogers pointed out that internet addiction is a kind of behavior like pathological gambling (Young & Rogers, 1998).’ ‘Negative feelings about real communication will undoubtedly prompt teenagers to avoid real life communication.’ Etc. Avoid such repetitions. ‘teenagers with depression or anxiety are more likely to alleviate their depression and anxiety’ – poorly constructed, repetitive. ‘In fact, many studies have confirmed the hypothesis above that there is a significant positive relationship between depression, anxiety and internet addiction. The higher the level of depression and anxiety is, the more serious the internet addiction is (Young, & Rogers, 1998).’ – but you cite only one source, a 24 years old one! Studies are either cross-sectional, not horizontal, or longitudinal. Horizontal study means to look at one age and compare the fundamental needs of a human during that specific time. Internet addiction and internet addiction are used alternatively. Hypotheses must be constructed based on more supporting sources, preferably as recent as possible. ‘Based on the literature at home and abroad, it is found that’ – remove this. Try and provide more references to support your ideas that are typically substantiated by only one source – and as recent as possible. E.g., here you use only a 21 years old source: ‘Researches in the field of pathological gambling have found out that women are worse at gambling than men (Tavares, Zilberman, Beites, & Gentil, 2001).’ ‘However, there are relatively few studies 84 on the direct discussion of internet addiction and obesity among teenagers (Kautiainen, Koivusilta, Lintonen, Virtanen, & Rimpelä, 2005).’ – I have just checked WoS: there are hundreds of articles published on this topic in the past 5 years. ‘internet addiction Test (IAT) developed by Young’ – year needed. You should compare your results with others in terms of concrete data for better research integrative value. ‘As teenagers who are open to new things, they are more likely to be interested in the internet.’ – poorly constructed. ‘The study found that the time spent online is positively correlated with internet addiction (Smahel, Brown, & Blinka, 2012),’ – should be ‘Smahel et al. (2012) found…’. ‘Therefore, it is necessary to improve their practical communication skills and abilities, and enhance their positive expectations for real communication through a series of training and learning. Finally, try to reduce teenagers' negative emotions (such as depression and anxiety).’ – who should do that? ‘getting twice the result with half the effort’ – remove this. ‘The rapid deterioration of girls’ – you mean their health or what? The conclusion, too short, should clarify the main contribution of the paper and the value added to the field. The manuscript requires major revisions to contextualize the merits of the study and potential uses of its methodology in future studies.
As the research is not related to hot emerging topics involving adolescents’ smartphone-based internet addiction and depression, anxiety, gender and weight, I suggest integrating such recent sources:
Lăzăroiu, G., Kovacova, M., Siekelova, A., and Vrbka, J. (2020). “Addictive Behavior of Problematic Smartphone Users: The Relationship between Depression, Anxiety, and Stress,” Review of Contemporary Philosophy 19: 50–56. doi:10.22381/RCP1920204
Green, M., Kovacova, M., and Valaskova, K. (2020). “Smartphone Addiction Risk, Depression Psychopathology, and Social Anxiety,” Analysis and Metaphysics 19: 52–58. doi:10.22381/AM1920205
Adams, C., Grecu, I., Grecu, G., and Balica, R. (2020). “Technology-related Behaviors and Attitudes: Compulsive Smartphone Usage, Stress, and Social Anxiety,” Review of Contemporary Philosophy 19: 71–77. doi:10.22381/RCP1920207
Kliestik, T., Scott, J., Musa, H., and Suler, P. (2020). “Addictive Smartphone Behavior, Anxiety Symptom Severity, and Depressive Stress,” Analysis and Metaphysics 19: 45–51. doi:10.22381/AM1920204
Reviewer 3 Report
While the topic of this paper is of interest, I have significant concerns with this paper. While the overall approach (using HLM to examine the relationship of depression and anxiety with internet addiction among adolescents) is appropriate, there are notable methodological issues. Additionally, the manuscript is difficult to read with inadequate explanation of key terms (e.g., equations on page 6), numerous instances of inappropriate word choice (e.g., “horizontal analysis”), and many grammatical errors (e.g., missing words). Furthermore, there is a mismatch of the references cited in-text and those listed at the end of the manuscript.
Methodological issues:
The sample design indicates that students were clustered in classrooms within schools. Was this clustering accounted for in your models?
Did you have any missing data? How did you handle it?
More details are required on the measures that were included in the student surveys. There are author names indicated in the text, but no matching references to examine to understand more about the scales and their psychometric properties.
The description of how obesity was measured is inadequate. There is (again) a reference indicated that is not included in the reference list. How was obesity measured? How do Chinese classification standards differ from, e.g., the US classification standards?
As noted in the limitations section, it is necessary to consider more individual factors. Do you have a measure of parental education? If so, why wasn’t this included?
The discussion section focuses on key measures that you did not include. For example, amount of time spent online is critical to control for. Was this measured in the surveys? If so, this should be included in the models. Additionally, for the obesity results, there is mention of categories of internet use. If this was measured, it should be included in the models. If not, then you can’t say that the results can be explained by something you didn’t measure.
The primary concern for me, however, is that it appears this is a study of adolescents who are typical internet users. For the IAT, scores of 31-49 are classified as "Mild: The child is an average online and screen user. He or she may surf the internet a bit too long at times, but seems to have control of screen usage." What proportion of the students met the criteria for internet addiction? An average decrease of 2 points from T1 to T2 and then an increase of 2 points from T2 to T3 suggests variation over time that stays within the “mild”/typical levels. I also want to know what proportion of the students met the criteria for depression and anxiety.